# Modelling and Determination of Parameters Influencing the Transfer of Microorganisms from Food Contact Materials

**DOI:** 10.3390/ijerph19052996

**Published:** 2022-03-04

**Authors:** Stephanie Maitz, Paul Jakob Schmid, Clemens Kittinger

**Affiliations:** Diagnostic & Research Institute of Hygiene, Microbiology and Environmental Medicine, Medical University of Graz, 8010 Graz, Austria; stephanie.maitz@medunigraz.at (S.M.); paul.schmid@medunigraz.at (P.J.S.)

**Keywords:** bacteria, transfer, incubation time, applied weights

## Abstract

The transfer of microorganisms on packaging materials to a contact surface has only been investigated in the context of laboratory-produced spiked packaging products and agar surfaces in small quantities (0.03–0.10%) so far. Correspondingly, this study focused on the localization of microorganisms on/in industrially produced packaging materials and on the establishment of an experimental laboratory set-up to determine and quantify the parameters influencing the microbial transport from surfaces and different layers of packaging materials to contact agar media. We established a simple model to determine the transfer of microorganisms from packaging materials to microbiological agar plates. In order to clarify the transfer of microorganisms within the material, the samples were split horizontally in their z-dimension, and so produced layers (inner layers) were investigated for their microbial transfer. The parameters incubation time, applied weight and bacterial load for the samples were investigated in more detail in the outer layers (front/back) and the inner layers. No significant difference in the microbial transfer was observed between the outer and inner layers of all samples. We indicate a time-dependent transfer to the media and an independence of the transfer from the applied weight. Moreover, the number of transferred microorganisms is not dependent on the bacterial load of the samples.

## 1. Introduction

The increase in packaging consumption has pushed manufacturers to optimize the essential requirements for their products [1]. This is an important issue in the food and packaging industry in relation to the application of materials being either directly or likely in contact with food. This includes materials used in the kitchen, for manufacturing food packaging materials and for food itself [2]. The manufacturers of packaging materials aim to provide cost-effective packaging that meets industry requirements and consumer demands, maintains the safety of packaged goods and reduces environmental impact. Hygienic aspects, gas permeability and the prolonged shelf life of stored goods are additional, equally important functions [3]. In the same way as the food industry, the packaging industry is subject to special regulations. The Regulation (EC) No. 1935/2004 states that packaging materials coming in contact with food must not transfer any of their ingredients into food in concentrations that could either endanger human health or change the food composition in an unsuitable way [4]. Due to these demands, the industry has particularly focused on the research on the physical and chemical properties of food packaging. However, the microbiological integrity of packaging material plays an important role as well [5,6]. A great amount of knowledge about the microbial composition in packaging products is available. It is known that packaging material has a certain microbial load after production, ranging from 10^1^ to 10^5^ CFU/g [7,8]. Mainly aerobic thermophilic spore formers were identified in those materials after the final production step [6,9]. Detailed studies about the residence of microorganisms on and/or in packaging products have not been sufficiently reported until now. The occurrence of microorganisms is not exclusively limited to the surface of packaging materials, but they can also be found in the matrix. Confocal laser scanning microscopy (CLSM) observations of polyethylene, mineral pigment or a biodegradable polymer-coated and high-density packaging materials revealed an uneven spatial distribution of microorganisms in optical sections in the z-direction [10]. However, besides the observations of microbial distribution in a complex three-dimensional network of packaging materials, there is still a significant lack of knowledge about how microorganisms are transferred from this network to a contact surface. Therefore, this study aimed to increase the knowledge about the microbial transfer in packaging materials in the different layers (front, back, inner and outer layers) and its impact on the transfer of microorganisms to target media. So far, food industries have demonstrated, in various studies, the successful transfer of microorganisms between hands and gloves to food and other surfaces involved in food production, storage and preparation [11,12,13]. There are already studies that have investigated and compared the microbial transfer of different spiked packaging materials (mainly cardboards and plastics) to packaged fruit. These studies revealed that the use of cardboard, compared to plastic, can significantly reduce the potential for the cross-contamination of packaged food [14,15,16]. Furthermore, the transfer of microorganisms from laboratory-produced packaging material spiked with labelled spores to agar plates in very low quantities (0.03–0.10%) has been reported [17]. However, to our knowledge, no experimental set-up has been developed to describe the transfer of microorganisms to a contact surface for industrially produced packaging material.

### Aim of the Study

This study focused on the establishment of a simple laboratory model to compare the microbial transfer from the outer and inner layers of a packaging material to contact media. On the basis of the known microbial load of the packaging material (determined according to ISO 4833-1:2013 [18]), we developed a protocol for transfer processes from the samples to a recipient agar to determine the dependence of the transferred microorganisms on incubation time and application weights.

## 2. Materials and Methods

### 2.1. Materials Studied

This study included six samples belonging to fiber-based food packaging materials (FCM) produced from different factories in Europe. The textures of each sample and chemical recipe were predefined by the factory at hand. Sampling was conducted under sterile conditions. The samples were individually wrapped in aluminum foil and packed in sealed sterile plastic bags directly after manufacturing by instructed workers and then sent to the laboratory. The samples were stored at room temperature. All packaging material samples were coded with a sequential number. From all samples, the total number of colony-forming units per 100 square centimeters (CFU/100 cm^2^) was determined. The samples belong to the food packaging sector, including the primary and secondary food packaging used in various applications (Table 1).

### 2.2. Determination of the Total Number of Colony-Forming Units per 100 Square Centimeters (CFU/100 cm^2^) for Each Sample

Sample homogenization was carried out according to ISO 4833-1:2013 [18]. Therefore, 1 g of native sample was weighed into a sterile plastic bag (VWR International, PA, USA) and 99 mL of sterile buffered peptone water (Carl Roth, Karlsruhe, Germany) was added. By visual inspection, the sample was homogenized using the BagMixer 400 SW (Interscience, Saint Nom la Bretèche, France) for 4 to 15 min, until the sample was completely dispersed and a homogenous suspension was obtained. Each sample was tested in duplicates (Figure 1). Colony counting was carried out according to ISO 4833-1:2013 [18]. After homogenization, the sample suspension was diluted (10^−1^–10^−3^) in buffered peptone water. Then, 1 mL of the suspension and of each dilution were transferred in a sterile Petri dish (90 mm × 16.2 mm, Fisher Scientific, Schwerte, Germany) and 15 mL tryptic soy agar (TSA, Oxoid, Munich, Germany) was added. In total, five plates for each dilution were made and incubated at 30 °C for 72 h (Figure 1). After incubation, the grown colonies on the agar were counted and the total number of colony-forming units per 100 square centimeters (CFU/100 cm^2^) was calculated for each sample considering the dilution and the replicates, according to the following formula:$$\mathrm{CFU}/100{\;\mathrm{cm}}^{2}=\mathrm{arithmetic}\;\mathrm{mean}\;\mathrm{of}\;\mathrm{counted}\;\mathrm{colonies}\;\times 1.1\times \;\mathrm{grammage}\;\mathrm{of}\;\mathrm{each}\;\mathrm{sample}\;\left(g/{\mathrm{cm}}^{2}\right)\times 100\;$$

An internal validation determined a recovery rate of colonies of 90%. Therefore, the arithmetic mean is multiplied by 1.1. The total CFU/100 cm^2^ for each sample is listed in Table 1.

### 2.3. Transfer Evaluation of Microorganisms to Agar Plates under Increasing Incubation Time, Increasing Applied Weight and from Outer and Inner Layers from Tested Samples

From each sample, a circle with an area of 50 cm^2^ was cut out. This area corresponds to the area of an agar plate. The outer layers of each sample were placed face down on tryptic soy agar plates (TSA, VWR, Leuven, Belgium). Three different experimental settings were realized to assess the impact of the parameters on the transfer of microorganisms (Figure 2).

First, the packaging material sample was exposed to different applied weights, including 100 g, 150 g, 200 g or 250 g, for 30 min (Figure 2(1)). Second, the samples were loaded with a weight of 250 g for different durations comprising 0 min, 5 min, 10 min, 15 min, 20 min and 30 min (Figure 2(2)). Tests with different incubation times and applied weights were performed eight times. Both experiments were tested on the outer layers of the samples, defined as front and back, to determine a possible difference in the transfer of microorganisms between them.

Third, the impact of the location of the microorganisms within the cross-section of the sample sheets was assessed. The outer layers of each sample were weighted with 250 g for 5 min on TSA plates. In addition, the packaging materials were split following the laminating splitting method of Knotzer et al. (2003) [19]. This established method in packaging material technology is used to investigate, for example, the internal fiber structure in packaging materials [20]. For the present purpose, we deviated from Knotzer et al. (2003). by omitting the step of staining the sample and by using a tape rather than employing a laminating foil for the splitting process. The latter prevented an additional heating step. The sample was split once approximately in the middle. Both halves of the obtained inner layers were loaded with 250 g for 5 min with the inner surface face down on the TSA. Splitting tests were repeated five times (Figure 2(3)).

After each testing, the agar plates were incubated at 37 °C for 24 h (Figure 2).

### 2.4. Data Analysis

Results were expressed in CFU/100 cm^2^. Statistical analyses were performed using GraphPad Prism 9 (GraphPad Software, San Diego, CA, USA). First, it was determined whether there was a significant difference in the transfer of microorganisms between front and back. All results of the incubation time and applied weight experiments were analyzed together using the Mann–Whitney U test. In addition, the evaluations of the individual samples for parameter settings of weights and time were summarized to measure the dependence of increasing applied weight and incubation times on microbial transfer using a locally weighted scatterplot smoothing (LOWESS), the Kruskal–Wallis test and the Mann–Whitney U test. All data from the outer/inner layer experiments were also initially summarized, and a difference in the microbial transfer was determined using the Mann–Whitney U test. However, all data from the parameter setting were not only evaluated collectively, but each sample was also analyzed individually using the Mann–Whitney U test and the Kruskal–Wallis test. A line was drawn through the mean value of all measured values for the incubation time and application weights in order to illustrate the dependence of the transfer on the tested parameters.

Subsequently, the number of all colonies transferred after 30 min to the TSA plates weighted with 250 g were determined and compared to the total number of CFU/100 cm^2^ of the samples. Furthermore, the transfer ratios per 100 square centimeters (%) were calculated using the following formula:$$\mathrm{Transfer}\;\mathrm{ratio}/100{\;\mathrm{cm}}^{2}\;\left(\%\right)=\frac{\mathrm{CFU}/100{\;\mathrm{cm}}^{2}\;\mathrm{recovered}\;\mathrm{on}\;\mathrm{the}\;\mathrm{agar}\;\mathrm{plates}\;\times 100}{\mathrm{CFU}/100{\;\mathrm{cm}}^{2}}$$

Results are expressed in CFU/100 cm^2^ with corresponding mean values ± respective standard deviation (SD). A *p*-value < 0.05 was defined as statistically significant.

## 3. Results

### 3.1. Subsection

#### 3.1.1. Characteristics of Tested Samples

For the six different packaging material samples, the number of colony-forming units per 100 square centimeters (CFU/100 cm^2^) ranged widely from 1.00 × 10^1^ to 8.63 × 10^5^ CFU/100 cm^2^ (Table 1).

#### 3.1.2. Transfer Analysis Comparing Front and Back of Tested Samples

To assess the different microbial transfer of six packaging materials between the front and back, the samples were tested on both outer layers in the experiments with the applied weight and incubation time. When comparing all data together, no significant difference (*p* = 0.51) was calculated between the front and the back (Figure 3A). However, when the data were compared for each sample individually, a significant difference (*p* < 0.01) between the two outer layers was calculated for sample 2 (Figure 3C). The results of samples 1 and 3–6 showed no significant difference (*p* = 0.06–0.94) between the tested outer layers (Figure 3B,D–G). Therefore, the results of the front and back of the tested parameters of applied weight and incubation time were presented separately for sample 2, while the data between the outer layers for samples 1 and 3–6 were summarized. A detailed overview of the transferred CFU/100 cm^2^ and the corresponding transfer ratios (%) are listed in Table 2.

#### 3.1.3. Transfer Analysis with Different Applied Weights

The transfer of microorganisms from the total number of CFU/100 cm^2^ of the samples to TSA plates from six investigated samples applied with increasing weights over a defined time of 30 min was investigated in more detail (Figure 2(1)). The transferred CFU/100 cm^2^ values of the applied weights were calculated for the outer layers (front and back) of all samples and for each sample individually. Results showed no significantly (*p* > 0.99) increased transfer of microorganisms from packaging materials to TSA with increased applied weight on the outer layers of tested FCM (Figure 4A). Moreover, the samples were analyzed separately. Again, no significant increase (*p* > 0.05) in microbial transfers with increasing applied weight could be determined (Figure 4B–G). Therefore, the microbial transfer is independent of the applied weight.

In addition, the packaging materials were divided into materials with a high number of CFU/100 cm^2^ (samples 1, 2 and 3) and a low number of CFU/100 cm^2^ (samples 4, 5 and 6). Samples with a high bacterial number for CFU/100 cm^2^ showed low transfer ratios (0.02–0.31%) compared to samples with a low CFU/100 cm^2^, which had higher transfer ratios (0.60–2.21%) of the outer layers. In contrast, samples with a low number of CFU/100 cm^2^ showed a lower transfer of microorganisms (1–4 CFU/100 cm^2^) to the TSA plates than samples with a high number of CFU/100 cm^2^ (8.3 × 10^1^–1.16 × 10^3^ CFU/100 cm^2^). Moreover, the SD between the applied weights was high for samples with a low number of CFU/100 cm^2^ (0.87–4.49) than for those with a high number of CFU/100 cm^2^ (<0.01–0.05, Table 3).

#### 3.1.4. Transfer Analysis with Increasing Incubation Time

The transfer of microorganisms from the total number of CFU/100 cm^2^ of the samples to TSA plates from six investigated samples weighted with 250 g was examined with increasing time (Figure 2(2)). The transfer was calculated for the outer layers (front and back) of all samples and each sample individually. The transfer of microorganisms from tested samples to the TSA plates appeared to be significantly (*p* = 0.02) time-dependent (Figure 5A). However, a detailed look at the evaluations of the individual samples according to their total CFU/100 cm^2^ classification revealed that samples with a high total CFU/100 cm^2^ (1, 2 and 3) showed a time-dependent transfer (*p* < 0.01, Figure 5B–D), while samples 4 and 6, with a low total CFU/100 cm^2^, showed no significant microbial transfer (*p* = 0.83–0.1, Figure 5E,G). On the contrary, sample 5, with a low total CFU/100 cm^2^, showed a time-dependent transfer (*p* = 0.02, Figure 5F).

The transfer ratios of selected incubation times were calculated for the outer layers of each sample. Packaging materials with a low number of CFU/100 cm^2^ showed a low number of transferred microorganisms (1–2 × 10^0^ CFU/100 cm^2^) to the TSA plate but high transfer ratios (0.05–1.54%) after 30 min compared to the samples with a high number of CFU/100 cm^2^ (<0.01–0.33%) (Table 4).

#### 3.1.5. Transfer Analysis of Inner and Outer Layers of the Samples

The transfer of bacteria from the outer and inner layers of six investigated samples of TSA plates were compared with each other. For this purpose, each sample was split horizontally once to assess the transfer of microorganisms from the inner layers (Figure 2(3)). The microorganisms were not only present at the outer layers of a sample, but also at the inner layers. The evaluations of all samples summarized together revealed no significant (*p* = 0.09) difference between the microbial transfer of inner layers and outer layers (Figure 6A). The analysis of each sample individually showed a significantly different transfer of microorganisms between the outer and inner layers of samples with a high total CFU/100 cm^2^ (*p* < 0.01, Figure 6B–D). In contrast, samples 4 and 5, with a low total CFU/100 cm^2^, showed no significant difference in the microbial transfer of tested layers (*p* > 0.05, Figure 6E,F). However, sample 6, with a low total CFU/100 cm^2^, showed a significant difference in the microbial transfer of tested layers to the TSA plates (*p* = 0.01, Figure 6G).

Packaging materials with a high number of CFU/100 cm^2^ showed higher transfer ratios from the inner layers (0.03–0.17%) compared to the outer layers (<0.01–0.03%). In contrast, sample 4 and 5, with a low number of CFU/100 cm^2^, showed no different transfer ratios between inner (<0.01–0.18%) and outer (<0.01–0.32%) layers. However, sample 6 presented a significant (*p* = 0.01) difference for the bacterial transfer between tested layers with a higher transfer ratio to the outer layers (2.36%) than to the inner layers (<0.01%, Table 5).

#### 3.1.6. Transfer Analysis Considering Transfer Ratios and the Total Number of CFU/100 cm^2^ of Tested Samples

The transfer of microorganisms from six investigated samples with increasing numbers of CFU/100 cm^2^ weighted with 250 g for 30 min on TSA plates were compared with each other. The number of transferred microorganisms is not associated with the total bacterial loading of the samples, e.g., the packaging material with the highest loading did not show the highest transfer ratio, and vice versa. Among the samples with a high number of CFU/100 cm^2^, there are neither indications of a constant transfer ratio nor of transfer ratios that increase linearly with the increasing number of CFU/100 cm^2^ of the samples. Sample 1, with the highest number of CFU/100 cm^2^, transferred the highest number of microorganisms (1.10 × 10^3^ CFU/100 cm^2^) to the TSA plates, but it did not demonstrate the highest transfer ratio (0.13%) compared to the other samples with high numbers of CFU/100 cm^2^. The packaging material with the second-highest number of CFU/100 cm^2^ (sample 2) transferred a lower number of microorganisms to the TSA plates than sample 1 (1.30 × 10^2^ CFU/100 cm^2^) and had the lowest transfer ratio (0.02%) compared to all tested samples. Sample 3 transferred a higher number of microorganism to TSA plates than sample 2, but less than sample 1 (4.37 × 10^2^ CFU/100 cm^2^). This packaging material showed the highest transfer ratios (0.33%) compared to all samples with a high number of CFU/100 cm^2^. The transfer ratios for samples with a low number of CFU/100 cm^2^ were more difficult to analyze and calculate. The transferred microorganisms were lower compared to samples with a high number of CFU/100 cm^2^ (<1 × 10^1^–6 × 10^0^ CFU/100 cm^2^). Nevertheless, the transfer ratios for samples 4–6 (0.32–1.06%) were higher than for samples 1–3 (0.02–0.33%) with a high bacterial loading (Figure 7, Table 6).

## 4. Discussion

This study compares the microbial transfer rates from outer layers (front and back) tested under different conditions and inner layers of a packaging material to contact media. Therefore, an adequate transfer model was established where factors can be analyzed in a controlled way to determine the dependence of the transferred microorganisms on incubation time and application weights. These parameters are of particular interest because reports from the food industry have determined that they are influencing factors with respect to the migration of microorganisms [21].

The experiments performed in this study are fundamentally different from those conducted already with food. Our experiments are designed to analyze the microbial transfer from and through the matrix of packaging materials. The first results demonstrated that spores remain in their loose position until they are mobilized by favorable conditions (e.g., liquids). An increased migration of microorganisms was observed especially from samples with a high number of CFU/100 cm^2^ (10^5^ CFU/100 cm^2^). In contrast, samples with a low number of CFU/100 cm^2^ (10^2^–10^1^ CFU/100 cm^2^) showed less transference. Our experiments indicate a more likely transfer of microorganisms from samples with a high bacterial load to the wet substrate.

The first question that arises when applying a packaging material is whether there is a difference between the two outer layers, defined in this study as front and back. Therefore, we tested all our samples on both outer layers first and compared the results with each other. To our knowledge, a difference between the transfer of microorganisms from the outer layers of a packaging material has not yet been tested. However, it could be that one layer reduces the transmission of microorganisms compared to the other layer. Our findings revealed that it makes a difference whether the evaluated data of all samples are analyzed together or each sample individually. While the results of all samples together showed no significant difference between front and back (*p* = 0.51), the analysis of the individual samples showed a significantly (*p* < 0.01) higher microbial transfer from the front than from the back for sample 2. This result suggests that microbial transfer depends on the properties of the sample variety (e.g., recycling stage, porosity, chemical additives, surface coatings, contact angle, etc.). Therefore, the results of front and back of tested parameters were depicted separately for this sample.

The influence of applied weight and its positive effect on increased bacterial transfer has been observed in the food industry [21,22]. The increased transfer through increased applied weight is based on the mechanism of bacterial adhesion. With increasing applied weight, the distances between the surfaces are reduced, the initial repulsive forces are avoided and binding forces and more specific interactions are promoted, which results in the transfer of microorganisms [23]. Despite these results, our analysis of all samples revealed that the transfer does not change significantly with increasing applied weight (Figure 4A). Even when comparing the transfer of microorganism of each sample, the bacterial transfer did not increase significantly with the increasing applied weight (Figure 4B–G, Table 3). These results indicate the same influence of low and high applied weight tested in this study, as they may give the same reduced distances between packaging materials and target medium, which leads to possible interactions and transmission between the surfaces and bacteria.

Until now, little was known about the influence of incubation time on the transfer of microorganisms to food [24]. A transfer over eight hours has shown a proportional relationship between the increase in the incubation time and the increase in the number of transferred microorganisms, according to Dawson et al. (2007) [25]. These results could also be observed in this study for evaluations on all samples and, especially, for samples with a high number of CFU/100 cm^2^. For those samples, the transfer appears to be significantly (*p* < 0.01) dependent on the incubation time, even in short time intervals until 30 min. The transfer of microorganisms after 30 min was significantly higher than the transfer after 5 min of contact (Table 4).

Moreover, the influence of the 3D effect on the transfer was studied in more detail. Therefore, the samples were split into two halves. We observed not only a presence of microorganisms on the outer layers of the samples, but also on their inner layers (matrix). The transfer of microorganisms from the packaging material was not significantly (*p* < 0.09) increased from the inner compared to the outer layers when incorporating all samples (Figure 6A). In contrast, packaging materials with a high microbial load transferred a significantly (*p* < 0.01) higher number of CFU/100 cm^2^ from their inner layers than from their outer layers. The significant transfer of bacteria and spores from the inner layers for samples with a low number of CFU/100 cm^2^ was significant only for sample 6 (*p* = 0.01), compared to their outer layers. Sample 6 has the lowest number of CFU/100 cm^2^ (1.00 × 10^1^ CFU/100 cm^2^) in this experiment. This means that, in the best case, no more than one colony can be transferred from the sample to the TSA plate. Therefore, there are only two scenarios for observing the transfer of this sample: one colony or none. This, in turn, explains the significant difference between the two layers. Nevertheless, the different transfer rate between the outer and inner layers is proven to be significant. These results are contrary to the observations of Suominen et al. (1997) [10]. In their study, the uneven presence of spores in the matrix of coated packaging materials was documented using confocal laser microscopy (CLSM). While Suominen et al. (1997) placed nutrients and water on the samples and observed the microbial growth via CLSM, we split the sample and applied them directly to the agar to analyze the growth of microorganisms. Our process favors the absorption of sufficient water available for the transfer of the microorganisms, contained in the sample matrix, to the contact material. The results demonstrated a transfer of microorganisms from both the outer layers and the matrix of the sample, using water as a transport medium. Similar to our study, Jaako et al. (2009) mainly demonstrated a spore transfer from the outer layers, but without splitting the samples [17]. According to our results, 0.02–0.11% of microorganisms of the samples with a high number of CFU/100 cm^2^ (10^5^ CFU/100 cm^2^) were transferred from the sample matrix when the samples were soaked with water. The assertion that only a very small number of microorganisms can be transferred from the matrix of the sample after wetting can be supported with our experimental set-up as well. Nevertheless, we assume that the spores are in a loose and mobile condition inside the matrix and can, therefore, be mobilized.

In this study, we compared the total number of CFU/100 cm^2^ of each sample (determined according to ISO 4833-1:2013 [18]) and the transferred microorganisms per 100 cm^2^ to a contact surface. Furthermore, the transfer ratios (%) were determined to assess a relationship between the microbial load and the transfer to an agar medium. The findings revealed that the transferred microorganisms (%) were not related to the bacterial load of the samples (Figure 7). First, we assumed a correlation between a high bacterial loading of the samples and a high transfer ratio of microorganisms to the contact material, but no considerable increase in the transfer ratio with increased bacterial loading of the packaging materials was observed with the developed transfer model. In more detail, sample 1, with the highest bacterial load, transferred the highest number of microorganisms to an agar plate, but calculated to its total number of CFU/100 cm^2^, it transferred only 0.13%. In comparison, sample 6, with the lowest bacterial load, transferred the smallest number of microorganisms, but this was 1.06% of the total number of CFU/100 cm^2^. According to these results, the transfer of microorganisms is not only influenced by the total number of CFU/100 cm^2^ of the analyzed samples. This strongly indicates the profound influence of physical properties (e.g., fiber type, porosity, mechanical strength, etc.) on the transfer. Therefore, a packaging material’s specific parameters and their correlations on the transfer ratios should be tested in further studies.

In summary, the increase in the transfer rate is time-dependent, but independent of the applied weight, or of the bacterial loading of the tested samples. According to the calculations of our study, between <0.01 and 2.48% microorganisms were transferred from packaging materials with a total bacterial load of 8.63 × 10^5^ CFU/100 cm^2^ to 1.00 × 10^1^ CFU/100 cm^2^ to a contact surface. Jaakko et al. (2009) observed also a very low transfer of spores of one species (*Bacillus thuringiensis*) to food (rice and chocolate), at between <0.01% and 0.03%, from highly spiked samples (with 10^8^ to 10^10^; viable *B. thuringiensis* spores) produced in the laboratory [17]. A recently published study reported that the surface of packaging materials with a high wettability (like the tested samples in this study) reduces the adhesion of the spores to the contact surface, which in turn reduces the transfer of spores to food [16,26]. This could explain the low transfer ratios determined in this study. In our experiment, the availability of water seems to facilitate the transfer to a contact surface. The samples tested have a low water activity. Therefore, the experimental model was developed to ensure water absorption of the packaging material from the agar plate (suction direction upwards). In order to allow a microbial transfer, the applied weight presses against the suction direction and the spores can be transferred to the agar. Further investigations on specific factors inherent to the packaging material will show their influences on the transfer (e.g., recycling stages, porosity, mechanical strength, etc.). In addition, a more detailed analysis of the material structure is of great interest for determining its transfer properties.

## 5. Conclusions

This study focuses on the invention of an experimental laboratory set-up to determine and quantify parameters that promote or inhibit microbial transport from the surfaces and matrix of packaging materials to microbiological agar plates. First, the outer layers, including the front and back of six different packaging samples, were investigated in more detail. No significant difference in the microbial transfer was observed when comparing those layers of all samples. When the results of each sample were analyzed individually, a significant difference in the transfer of microorganisms between the front and the back of one sample was detected. Second, all samples were split horizontally in their z-dimension to determine the transfer of microorganisms within the matrix (inner layers) to a contact agar. Again, no significant difference was measured between outer and inner layers when comparing all samples. The analysis of each sample individually revealed a significant difference in the transfer of microorganisms between those layers of four samples. Third, we focused on the parameters of incubation time, applied weight and bacterial load of the samples to test their impact on the transfer to media used in microbiology. Results revealed a time-dependent and an applied weight-independent transfer to the contact media. The bacterial load of the samples had no influence on the number of transferred microorganisms. Moreover, a transfer ratio between <0.01 and 2.48% of packaging materials with a total bacterial load of 8.63 × 10^5^ CFU/100 cm^2^ to 1.00 × 10^1^ CFU/100 cm^2^ was determined. Our study strongly indicates that the properties of the packaging material itself profoundly influence the transfer.

## Figures and Tables

**Figure 1 ijerph-19-02996-f001:**
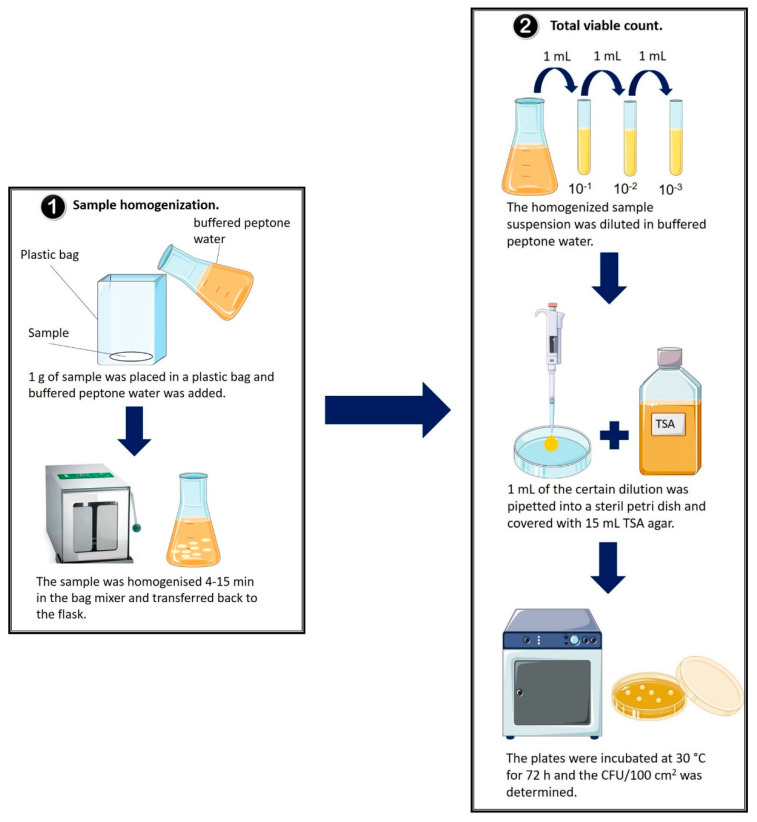
Laboratory experiments for sample homogenization (**1**) and measurement of the colony-forming units per 100 square centimeters (CFU/100 cm^2^) (**2**) from each sample. (**1**) To homogenize the sample in the bag mixer, 1 g sample was weighed into a sterile plastic bag and covered with 99 mL sterile buffered peptone water [18] (**2**) After homogenization, 1 mL of the suspension and dilution in buffered peptone water (10^−1^–10^−3^) were transferred to a sterile Petri dish, and 15 mL of tryptic soy agar (TSA) was added and incubated at 30 °C for 72 h. After incubation, the number of CFU/100 cm^2^ was calculated [18]. All tests were performed in duplicates.

**Figure 2 ijerph-19-02996-f002:**
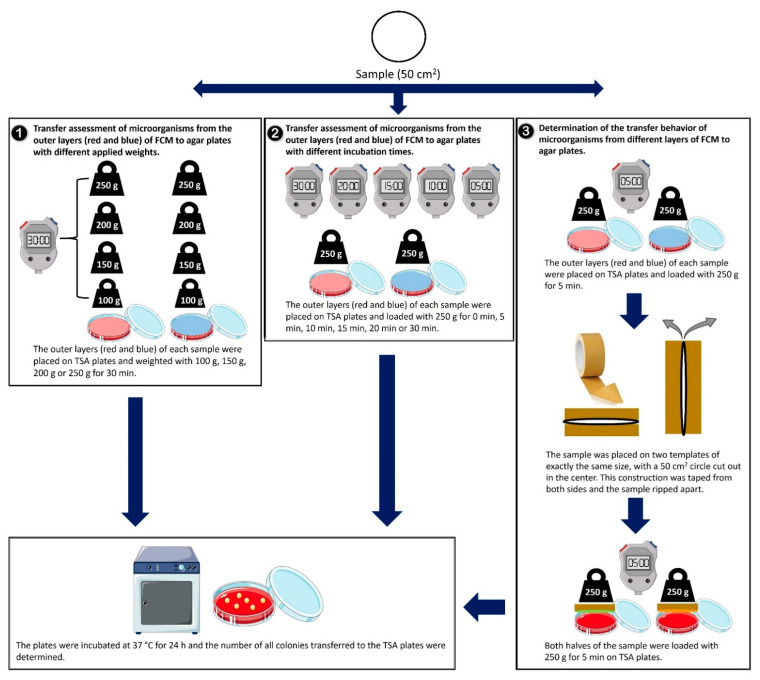
Experimental set-up to determine the transfer of microorganisms of food packaging materials (FCM) tested with different applied weights (**1**), increasing incubation time (**2**) and outer and inner layers (**3**). From each packaging material, a circle with an area of 50 cm^2^ was cut out. The outer layers (red: front, blue: back) of each sample were placed on tryptic soy agar (TSA) plates. (**1**) The transfer of microorganisms was measured by weighting the samples with 100 g, 150 g, 200 g and 250 g for 30 min. (**2**) The transfer of bacteria was measured by loading the samples with 250 g for 0 min, 5 min, 10 min, 15 min, 20 min and 30 min. (**3**) The transfer of microorganisms was not only observed from the outer (red and blue) but also from the inner layers (green and orange) of the sample. Therefore, the packaging material was split into two halves. The different layers were weighted with 250 g for 5 min on TSA plates. After the experimental setting of 1, 2 and 3, the agar plates were incubated at 37 °C for 24 h. Subsequently, the numbers of all colonies transferred to the TSA plates were determined. The weights and time experiments were performed eight times. Splitting tests were conducted five times.

**Figure 3 ijerph-19-02996-f003:**
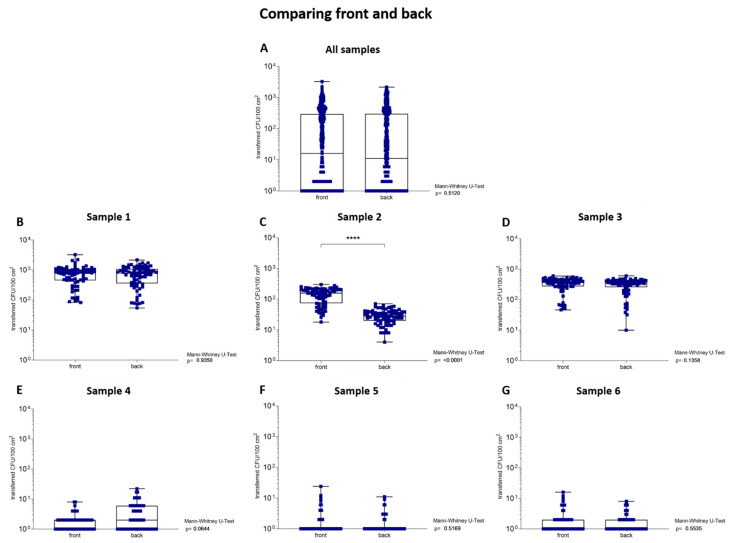
Comparison of the transfer of colony-forming units per 100 square centimeters (CFU/100 cm^2^) from the front and back of packaging materials to contact media. The samples are listed in increasing order. The nodes (blue) depict transferred CFU/100 cm^2^ from the outer layers of the packaging material to the TSA plates after the samples were loaded with 100 g, 150 g, 200 g and 250 g for 30 min, or loaded with 250 g for 5 min, 10 min, 15 min, 20 min and 30 min. A *p*-value below 0.05 was defined as statistically significant. (**A**) No significant difference was calculated between front and the back when summarizing all data (*p* < 0.01). (**C**) In contrast, when analyzing the data of each sample, a significant difference (*p* < 0.01) between front and back was calculated for sample 2. (**B**,**D**–**G**) The evaluations for samples 1 and 3–6 revealed no significant difference (*p* > 0.05) between both tested outer layers. **** indicates statistical significance with a *p*-value below 0.0001.

**Figure 4 ijerph-19-02996-f004:**
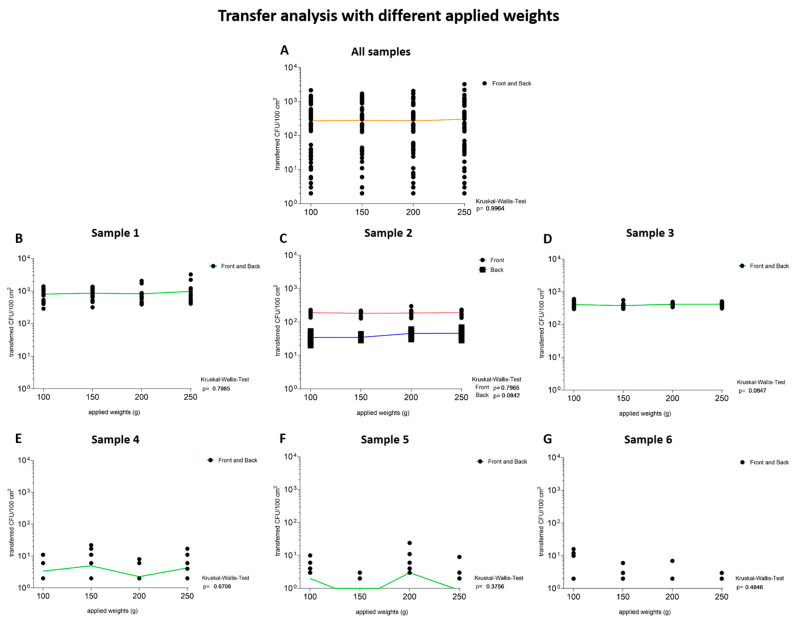
The transfer of colony-forming units per 100 square centimeters (CFU/100 cm^2^) from packaging materials to contact media is independent of the applied weight. The applied weights in grams are listed in increasing order. The nodes (black) depict transferred CFU/100 cm^2^ from the outer layers of the packaging material to the TSA plates after the samples were loaded with 100 g, 150 g, 200 g and 250 g for 30 min. The locally estimated scatterplot smoothing (LOWESS: orange) and the Kruskal–Wallis test were calculated to analyze the dependence on the applied weight of the transfer. Lines were drawn through the mean values of all measured data points to illustrate the dependence of the transfer on the tested parameter. Green lines illustrate data from both outer layers, while red lines represent data from the front and blue lines from the back. A *p*-value of below 0.05 was defined as statistically significant. With increasing applied weight, the transferred CFU/100 cm^2^ did not increase significantly (*p* > 0.05). Therefore, the transfer is independent of the applied weight in this experiment. (**A**–**G**) The graphs indicate no weight dependent transfer of microorganisms over all tested samples (*p* > 0.05).

**Figure 5 ijerph-19-02996-f005:**
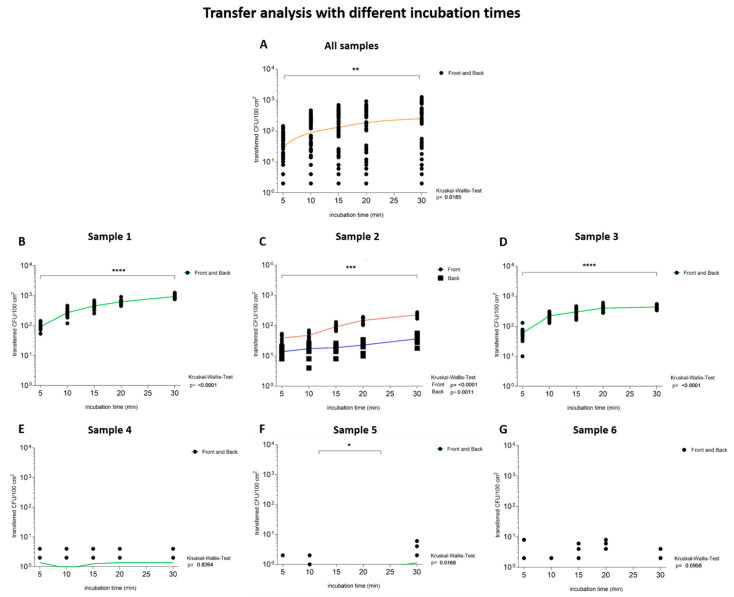
The transfer of microorganisms is time-dependent for samples with a high total CFU/100 cm^2^. The increasing incubation time in minutes is depicted. The nodes (black) depict transferred CFU/100 cm^2^ from the outer layers of the packaging material to the TSA plates after the samples were loaded with 250 g for 5 min, 10 min, 15 min, 20 min and 30 min. The locally estimated scatterplot smoothing (LOWESS: orange) and the Kruskal–Wallis test were calculated to analyze the dependence on incubation time of the transfer. Lines were drawn through the mean values of all measured data points to illustrate the dependence of the transfer on the tested parameter. Green lines illustrate data from both outer layers, while red lines represent data from the front and blue lines from the back. A *p*-value of below 0.05 was defined as statistically significant. (**A**) The graph indicates a time dependent transfer of microorganisms over all tested samples, since the transferred CFU/100 cm^2^ increases significantly (*p* = 0.02) with increasing incubation time. (**B**–**D**) Packaging materials with a high total CFU/100 cm^2^ showed a time-dependent transfer (*p* < 0.01). (**E**,**G**) The evaluations revealed no significant microbial transfer (*p* > 0.05) of samples with a low total CFU/100 cm^2^. (**F**) Sample 5, with a low total CFU/100 cm^2^, showed a time-dependent transfer (*p* = 0.02). * Indicates statistical significance with a *p*-value below 0.05, ** indicates statistical significance with a *p*-value below 0.01, *** indicates statistical significance with a *p*-value below 0.001 and **** indicates statistical significance with a *p*-value below 0.0001.

**Figure 6 ijerph-19-02996-f006:**
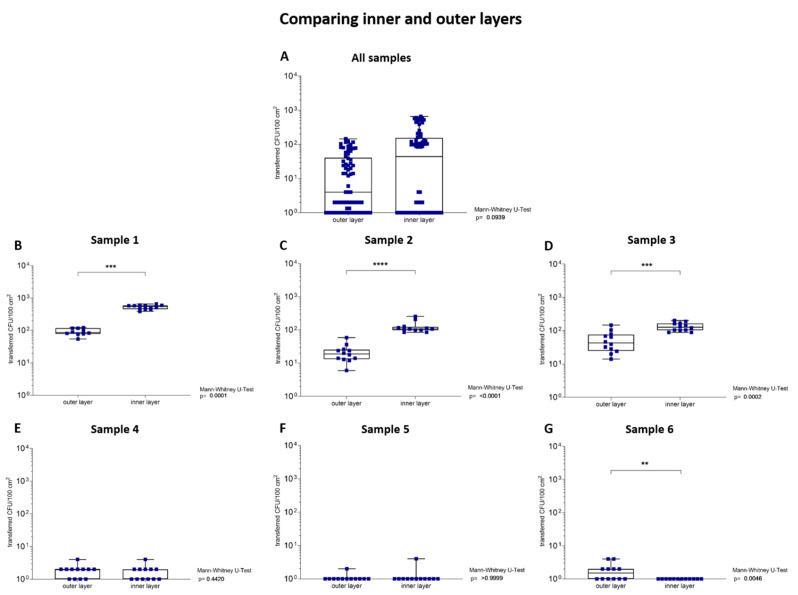
Significantly increased transfer of microorganisms from the inner layers of tested samples, compared to their outer layers. The nodes (blue) represent transferred microorganisms from the outer and inner layers of packaging material to TSA plates after the samples were loaded with 250 g for 5 min. The Mann–Whitney U test was used to measure whether there is a significant difference in the number transferred microorganisms between the outer and inner layers of the samples. A *p*-value of below 0.05 was defined as statistically significant. (**A**) The numbers of CFU/100 cm^2^ transferred from the inner layers to the agar plates were not significantly (*p* = 0.09) increased compared to the outer layers. (**B**–**D**) Samples with a high total CFU/100 cm^2^ showed a significantly different transfer between outer and inner layers (*p* < 0.01). (**E**,**F**) The evaluations revealed no significant microbial transfer (*p* > 0.05) of samples 4 and 5. (**G**) Packaging materials 6 showed a significantly different transfer between outer and inner layers (*p* = 0.01). ** Indicates statistical significance with a *p*-value below 0.01, *** indicates statistical significance with a *p*-value below 0.001 and **** indicates statistical significance with a *p*-value below 0.0001.

**Figure 7 ijerph-19-02996-f007:**
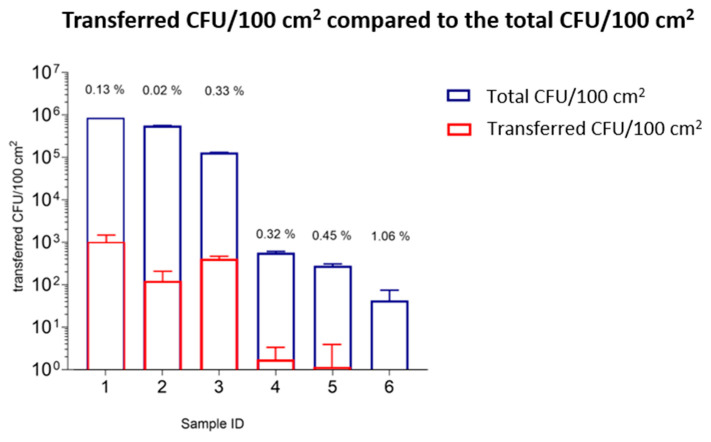
The number of transferred microorganisms to a contact surface is not related to the total bacterial loading of the samples. The samples are listed with respect to the decreasing bacterial loading (CFU/100 cm^2^). The blue bars present the total number of CFU/100 cm^2^ of each sample. Red bars depict the transferred microorganism from the outer layers of the packaging material to the TSA plates, after the samples were loaded with 250 g for 30 min. For each sample, the transfer ratio (%) was calculated. Sample 1, with the highest number of CFU/100 cm^2^ (8.63 × 10^5^ CFU/100 cm^2^), did not demonstrate the highest transfer ratio (0.13%) compared to the other tested samples. In contrast, the sample with the lowest bacterial loading, sample 6 (6.49 × 10^1^ CFU/100 cm^2^), showed the highest transfer ratio (1.06%). Therefore, the transfer ratio was not associated with the total bacterial loading of tested samples.

**Table 1 ijerph-19-02996-t001:** Packaging material samples and corresponding food applications. All tested samples were coded with a sequential number. The total numbers of colony-forming units per 100 square centimeters (CFU/100 cm^2^) were measured according to ISO 4833-1:2013 [18]. The packaging applications are in accordance with the manufacturer’s specifications.

Classification of Samples	Sample ID	Total Number of CFU/100 cm^2^	Food Packaging Applications	g/cm^2^
Samples with high numbers of CFU/100 cm^2^	1	8.63 × 10^5^	secondary food packaging: dry, non-fatty, peeled or washed	9.60 × 10^−3^
	2	5.64 × 10^5^	secondary food packaging: dry, non-fatty, peeled or washed	1.34 × 10^−2^
	3	1.32 × 10^5^	dry, moist, fatty	1.10 × 10^−2^
Samples with low numbers of CFU/100 cm^2^	4	5.00 × 10^2^	dry, moist, fatty (baking up to 220 °C)	3.60 × 10^−3^
	5	3.00 × 10^2^	dry, moist, fatty	7.00 × 10^−3^
	6	1.00 × 10^1^	dry, moist, fatty	1.18 × 10^−2^

**Table 2 ijerph-19-02996-t002:** Determination of the microbial transfer from the front and back of tested samples to tryptone soy agar (TSA) plates. The samples are labeled with respect to their numbers of colony-forming units per 100 square centimeters (CFU/100 cm^2^) in descending order (1–6). Data are shown as mean ± SD. The transfer ratios are declared as a percentage (%). A *p*-value of <0.05 was defined as statistically significant.

			Sample Number	1	2	3	4	5	6
Transfer of microorganisms to TSA plates	outer layers		CFU/100 cm^2^	8.63 × 10^5^	5.64 × 10^5^	1.32 × 10^5^	5.44 × 10^2^	2.62 × 10^2^	6.49 × 10^1^
		front	transferred CFU/100 cm^2^	7.91 × 10^2^ ± 5.28 × 10^2^	1.19 × 10^2^ ± 8.07 × 10^1^	3.48 × 10^2^ ± 1.40 × 10^2^	1 × 10^0^ ± 2 × 10^0^	1 × 10^0^ ± 4 × 10^0^	2 × 10^0^ ± 3 × 10^0^
			transfer ratio (%)	0.09 ± <0.06	0.02 ± 0.01	0.26 ± 0.11	0.26 ± 0.32	0.48 ± 1.36	2.48 ± 4.55
		back	transferred CFU/100 cm^2^	7.69 × 10^2^ ± 4.64 × 10^2^	6.56 × 10^1^ ± 7.02 × 10^1^	3.35 × 10^2^ ± 1.43 × 10^2^	2 × 10^0^ ± 2 × 10^0^	1 × 10^0^ ± 1 × 10^0^	1 × 10^0^ ± 2 × 10^0^
			transfer ratio (%)	0.09 ± 0.05	0.01 ± 0.01	0.25 ± 0.11	0.30 ± 0.33	0.21 ± 0.57	1.75 ± 2.90

**Table 3 ijerph-19-02996-t003:** Transfer of microorganisms from samples to tryptone soy agar (TSA) plates tested with different applied weights. The samples are labeled with respect to their numbers of colony-forming units per 100 square centimeters (CFU/100 cm^2^) in descending order (1–6). The experimental set-up was performed as described in Figure 2(1). Data are shown as mean ± SD. The transfer ratios are declared as a percentage (%). A *p*-value of <0.05 was defined as statistically significant.

		Sample Number	1	2	3	4	5	6
Transfer of microorganisms to TSA plates	outer layers	CFU/100 cm^2^	8.63 × 10^5^	5.64 × 10^5^	1.32 × 10^5^	5.44 × 10^2^	2.62 × 10^2^	6.49 × 10^1^
		transferred CFU/100 cm^2^	1.16 × 10^3^ ± 4.31 × 10^2^	front: 1.45 × 10^2^ ± 6.33 × 10^0^ back: 8.3 × 10^1^ ± 3 × 10^0^	4.05 × 10^2^ ± 6.40 × 10^1^	4 × 10^0^ ± 5 × 10^0^	2 × 10^0^ ± 4 × 10^0^	1 × 10^0^ ± 3 × 10^0^
		transfer ratio (%)	0.13 ± 0.05	front: 0.03 ± <0.01 back: 0.02 ± <0.01	0.31 ± 0.05	0.67 ± 0.87	0.60 ± 1.44	2.12. ± 4.49
		Kruskal–Wallis test	0.80	front: 0.80 back: 0.08	0.08	0.67	0.38	0.48

**Table 4 ijerph-19-02996-t004:** Transfer of microorganisms from the packaging material samples to tryptone soy agar (TSA) plates with different tested incubation times. The samples are labeled with respect to their numbers of colony-forming units per 100 square centimeters (CFU/100 cm^2^) in descending order (1–6). The experimental set-up was performed as described in Figure 2(2). Data are shown as mean ± SD. The transfer ratios are declared as a percentage (%). A *p*-value of <0.05 was defined as statistically significant.

			Sample Number	1	2	3	4	5	6
Transfer of microorganisms to TSA plates	outer layers		CFU/100 cm^2^	8.63 × 10^5^	5.64 × 10^5^	1.32 × 10^5^	5.44 × 10^2^	2.62 × 10^2^	6.49 × 10^1^
		5 min	transferred CFU/100 cm^2^	9.30 × 10^1^ ± 2.20 × 10^1^	front: 3.90 × 10^1^ ± 1.00 × 10^1^ back: 1.40 × 10^1^ ± 4 × 10^0^	5.90 × 10^1^ ± 2.50 × 10^1^	1 × 10^0^ ± 2 × 10^0^	<1 × 10^0^ ± <1 × 10^0^	1 × 10^0^ ± 2 × 10^0^
			transfer ratio (%)	0.01 ± <0.01	front: 0.01 ± <0.01 back: <0.01 ± <0.01	0.04 ± <0.02	0.25 ± 0.31	0.05 ± 0.18	1.54 ± 3.08
		30 min	transferred CFU/100 cm^2^	9.30 × 10^2^ ± 1.70 × 10^2^	front: 1.65 × 10^2^ ± 9.73 × 10^1^ back: 9.55 × 10^1^ ± 8.15 × 10^1^	4.36 × 10^2^ ± 6.80 × 10^1^	1 × 10^0^ ± 1 × 10^0^	2 × 10^0^ ± 4 × 10^0^	1 × 10^0^ ± 1 × 10^0^
			transfer ratio (%)	0.11 ± 0.02	front: 0.03 ± 0.02 back: 0.02 ± <0.01	0.33 ± 0.05	0.25 ± 0.21	0.67 ± 1.35	0.96 ± 1.80
		comparing 5 min and 30 min	Kruskal–Wallis test	<0.01	front: <0.01 back: <0.01	<0.01	0.83	0.02	0.10

**Table 5 ijerph-19-02996-t005:** Determination of the transfer of microorganisms from the inner and outer layers of tested samples to tryptone soy agar (TSA) plates. The samples are labeled with respect to their numbers of colony-forming units per 100 square centimeters (CFU/100 cm^2^) in descending order (1–6). The experimental set-up was performed as described in Figure 2(3). Data are shown as mean ± SD. The transfer ratios are declared as a percentage (%). A *p*-value of <0.05 was defined as statistically significant.

			Sample Number	1	2	3	4	5	6
Transfer of microorganism to TSA plates	5 min		CFU/100 cm^2^	8.63 × 10^5^	5.64 × 10^5^	1.32 × 10^5^	5.44 × 10^2^	2.62 × 10^2^	6.49 × 10^1^
		outer layer	transferred CFU/100 cm^2^	9.10 × 10^1^ ± 2.06 × 10^1^	2.20 × 10^1^ ± 1.30 × 10^1^	5.50 × 10^1^ ± 3.80 × 10^1^	2 × 10^0^ ± 1 × 10^0^	<1 × 10^0^ ± 1 × 10^0^	1 × 10^0^ ± 1 × 10^0^
			transfer ratio (%)	0.01 ± <0.01	<0.01 ± <0.01	0.03 ± 0.02	0.32 ± 0.20	<0.01 ± <0.01	2.36 ± 2.30
		inner layer	transferred CFU/100 cm^2^	5.28 × 10^2^ ± 7.80 × 10^1^	1.25 × 10^2^ ± 5.00 × 10^1^	1.32 × 10^2^ ± 3.70 × 10^1^	1 × 10^0^ ± 1 × 10^0^	<1 × 10^0^ ± 1 × 10^0^	<1 × 10^0^ ± <1 × 10^0^
			transfer ratio (%)	0.06 ± 0.01	0.02 ± <0.01	0.11 ± 0.02	0.18 ± 0.26	<0.01 ± <0.01	<0.01 ± <0.01
		comparing layers	Mann–Whitney U test	<0.01	<0.01	<0.01	0.44	<0.99	<0.01

**Table 6 ijerph-19-02996-t006:** Transfer analysis considering transfer ratios and number of CFU/100 cm^2^ of tested samples to tryptone soy agar (TSA) plates. The samples are labeled with respect to their numbers of colony-forming units per 100 square centimeters (CFU/100 cm^2^) in descending order (1–6). The experimental set-up was performed as described in Figure 2(2). Data are shown as mean ± SD. The transfer ratios are declared as a percentage (%). A *p*-value of <0.05 was defined as statistically significant.

			Sample Number	1	2	3	4	5	6
Transfer of microorganisms to TSA plates			CFU/100 cm^2^	8.63 × 10^5^	5.64 × 10^5^	1.32 × 10^5^	5.44 × 10^2^	2.62 × 10^2^	6.49 × 10^1^
	30 min	outer layers	transferred CFU/100 cm^2^	1.10 × 10^3^ ± 4.80 × 10^2^	1.30 × 10^2^ ± 8.90 × 10^1^	4.37 × 10^2^ ± 6.20 × 10^1^	2 × 10^0^ ± 2 × 10^0^	2 × 10^0^ ± 3 × 10^0^	<1 × 10^1^ ± 1 × 10^1^
			transfer ratio (%)	0.13 ± 0.06	0.02 ± 0.02	0.33 ± 0.05	0.32 ± 0.30	0.45 ± 1.06	1.06 ± 1.82

## Data Availability

Data are contained in the article.

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
