# Peer review of "Modelling and Determination of Parameters Influencing the Transfer of Microorganisms from Food Contact Materials"

_ijerph, 2022, doi:10.3390/ijerph19052996_

Round 1
Reviewer 1 Report
It is an interesting study that shows the modeling and determination of parameters influencing the transfer of microorganisms from food contact materials, whose results could help improve the properties packaging materials.
The title allows the identification of the topic and contemplates the study variables in a clear and concise way.
The summary is concrete and evidences the structure of the article; likewise, it has a clear description of the objective, and exposes the method used in the investigation.
The keywords respond to the subject.
The introduction develops the background of the investigation, stating the purpose and objective of the investigation.
The methodology correctly exposes the procedures for the design of the investigation and for its analysis
The article presents coherence in the summary, introduction, objectives, methodology, and results, has a conceptual basis that supports the presented argument.
The results are clear and correspond to the proposed objectives, which are presented properly for easy understanding and derive directly from the analysis of the data collected, contributing to the solution of the problem that has been raised.
The conclusions presented provide knowledge about the properties of packaging and its effect on microbial transfer to food.
Comments
After a thorough review, I consider that the document is ready to be published
Set superscript 2 in cm throughout the manuscript
Figures 2-5 could be made larger so that it can be seen more clearly
Author Response
Thank you for your detailed comments and positive feedback.
Unfortunately, the 2 in cm can not be superscript in the formulas added. In the text it is superscripted.
In the published version, all figures can be enlarged by clicking on them.
Reviewer 2 Report
This paper on "Modelling and determination of parameters influencing the 2 transfer of microorganisms from food contact materials" is very meaningful in terms of food safety, however I have the following suggestions. 1. This paper recommends a more detailed explanation of the reasons for the selection of "packaging materials", because the difference of packaging materials is an important factor affecting the results, and the research results of other literatures should be compared. 2. In "5. Conclusions" (line 508), the differences between the research results of this paper and other literatures should be emphasized, and the academic value of the results of this paper should be explained.
Reviewer 3 Report
The manuscript presents relevant results for food safety and is within the scope of the journal. Also, the English language is well spelled with minor corrections to be done. You can find my comments and suggestions below:
- Abstract
- Line 17 to 18: I suggest you could change the sentence as “the parameters incubation time, applied weights and bacterial load of the samples were investigated in more detail in the outer layers (front and back) and the inner layers”
- Introduction
- Line 42: “…certain microbial load…”. Could you please specify the load and indicate an appropriate citation for that?
- Material and Methods
- Line 71: ISO 4833-1 citation should be presented in an uniformized and numbered form.
- Line 125 to 128: Knotzer et al. needs a citation number.
- Line 132: replace “teste” by “tests”.
- Line 137: Why did you use Microsoft Excel 2013 and not, for instance, SPSS or SAS?
- Results
- Lines 173: Replace “p = 0.94-0.06” by “p = 0.06-0.94”.
- Line 186: Indicate the exact p-value or p > 0.05.
- Line 196: What about the results for 4,5 and 6 samples. I suggest that you could add it in the text.
- Line 200: I suggest that you could delete this sentence “The accumulation of errors was higher for samples with lower CFU/100 cm2”
- Lines 206 to 208: This part should be in the Statistical analysis section, not here.
- Line 213: Replace “. While” by ”,while”.
- Line 233: Instead of p=0.44->0.5, replace it by > 0.5 or indicate the exact p-value after samples 4 and 5.
- Line 242: Replace “of” by “to” and add “to” after “than”.
- Line 279: Please correct the number after ISO citation.
- Lines 299, 317, 336, 360 and 380; Table 2, 3, 4, 5 and 6: Some SD are omitted in the Tables. I suppose it was due to a formatting problem. Please, correct it.
- Discussion
- The citations are not uniformized, since some of them have a number while others are in the format author and year. Please, correct it.
- Line 401: replace “analyses” by “analyse”.
- Line 420: Which properties could have influenced the distinct transfer of microorganisms for sample 2?
- Line 432: replace “few” by “low”.
- Line 437: replace “an” by “a”.
- Line 444 to 445: “The applied weights in combination with incubation time plays an important role in respect to transfer.” Please, discuss a lit bit more this aspect.
- Line 450: Correct to “than from the outer”
- Line 454: Discuss possible reasons for the different results obtained for sample 6.
- Line 497: Replace “reduce” by “reduces”.
Round 2
Reviewer 2 Report
The author's reply I accept in principle, the two suggestions I made during the first review. Although the authors have made some explanations, they still cannot fully explain these two concerns, the authors also did his best to make some corrections. As a result, the paper can now be accepted for publication as is.